# Impact of Exogenous Xylanase and Phytase, Individually or in Combination, on Performance, Digesta Viscosity and Carcass Characteristics in Broiler Birds Fed Wheat-Based Diets

**DOI:** 10.3390/ani13020278

**Published:** 2023-01-13

**Authors:** Urooj Anwar, Muhammad Riaz, Muhammad Farooq Khalid, Riaz Mustafa, Umar Farooq, Muhammad Ashraf, Hassan Munir, Muhammad Auon, Mubasher Hussain, Munawar Hussain, Muhammad Farhan Ayaz Chisti, Muhammad Qamar Bilal, Abd ur Rehman, Muhammad Aziz ur Rahman

**Affiliations:** 1Institute of Animal and Dairy Sciences, University of Agriculture, Faisalabad 38000, Pakistan; 2Sub-Campus Toba Tek Singh, University of Agriculture Faisalabad, Toba Tek Singh 36050, Pakistan; 3Department of Agronomy, University of Agriculture, Faisalabad 38000, Pakistan; 4Sind Feed and Allied Products, Karachi 75600, Pakistan; 5Department of Animal Sciences, College of Agriculture, University of Sargodha, Sargodha 40100, Pakistan

**Keywords:** storage, wheat, growth, nutrient, digesta viscosity, broilers

## Abstract

**Simple Summary:**

Wheat is a staple food in Pakistan and Government of Pakistan stores huge amount of wheat every year to maintain food supply chains especially during scarcity period. Every year, surplus stored wheat is provided to poultry feed manufacturers at lower price and poultry feed manufacturers use stored wheat in diet of poultry with different exogenous enzymes. However, no data is available on the utilization of stored wheat (1.5 and 2.5 years storage) with or without exogenous enzymes in broilers. This study will provide evidence that storage of wheat will require exogenous enzymes in the diet of broilers for enhanced productivity.

**Abstract:**

The current study was conducted to evaluate the effects of stored wheat-based diet (1.5 and 2.5 years stored wheat) with and without the supplementation of xylanase and phytase enzymes in combination or individually on performance parameters, digestibility, digesta viscosity and carcass characteristics of broilers. For this purpose, a total of 640-day-old male broilers were randomly distributed to the 64 pens, and each pen had 10 birds. Two basal isocaloric and isonitrogenous diets contained 1.5 and 2.5 years stored wheat were formulated in this experiment. In the current study, experimental feeds were prepared by supplementing exogenous enzymes in both basal diets with xylanase (500 XU), phytase (500 FTU) alone or in a combination of phytase and xylanase. Performance parameters data represents that both in starter phase and finisher phase, inclusion of exogenous enzymes xylanase and phytase in both basal diets alone or in combination enhance the feed intake, and body weight gain (*p* < 0.05) and improve the feed conversion ratio in overall phase (*p* < 0.05). Similarly, supplementation of exogenous xylanase and phytase alone or in combination enhance the nutrient digestibility and reduce the digesta viscosity (*p* < 0.05). Based on the results of this experiment, it is concluded that supplementation of exogenous xylanase and phytase enzymes alone or in combination in wheat-based diets (stored wheat 1.5 and 2.5 years) enhance and improves the performance of birds.

## 1. Introduction

A poultry diet mainly consists of plant-based ingredients that contain a number of anti-nutritional factors and phytate. Non-starch polysaccharides (NSPs) are the most common anti-nutritional factors present in plant-based ingredients diet. Nutritionally a significant amount of NSPs are found in various grains such as wheat, barley, oats and rye and in legume seeds while phytate is present in all plant-based ingredients. Phytate and NSPs negatively affect the digestion of nutrients and ultimately result in poor performance of birds [1,2,3,4].

Wheat is used as an energy source in the diet of broiler birds in many parts of the world. The nutritive value of wheat is lower due to the presence of soluble NSPs. Non starch polysaccharides are complex carbohydrates that resist hydrolysis by enzymes and impair the digestion of other nutrients as well. Wheat contain arabinoxylans as soluble NSPs [5] and bran and endosperm of wheat contains 33.3% and 64.2% of soluble arabinoxylans [6] that is responsible for increing the viscosity of digesta, lowering the rate of passage of feed, utilization of nutrients and performance of birds [7]. However, various exogenous enzymes such as xylanase, protease, phytase and beta-glucanases are mostly used in the diet of poultry birds for improving the utilization of nutrients such as energy and amino acids and for lowering the detrimental effects of NSPs, that resulting in increased performance of birds [8]. It has been reported that exogenous enzymes, especially carbohydrases hydrolyze the aleurone layer of cereals that comprises endosperm or starch and enhance the digestibility and utilization of nutrients [9]. Like many other vegetative sources, wheat also contains a binding form of phosphorus that is called phytate. Birds cannot use this bound form of phosphorus due to their limited ability to secrete enzyme phytase. In wheat, phytate molecules are encapsulated with proteins within cells of the aleurone layer and are surrounded by carbohydrates that affect nutrient accessibility [8]. To overcome the negative effects of phytate phosphorus, exogenous phytase is supplemented in the diet of broiler birds [10]. Iyayi et al. [11] reported that supplementation of phytase improved the performance of broiler birds in terms of growth performance because phytase increase available phosphorus.

Supplementation of exogenous enzymes such xylanase and phytase results in improved digestion and absorption of nutrients like starch and protein in the gastrointestinal tract of the poultry birds and lesser amount of undigested material reaching in the lower part of the gastrointestinal tract [12]. Better digestion and increase in absorption of nutrients result in favorable microflora in the gastrointestinal tract that result in improved status of gut health, intestinal morphology and overall good effect on the health of birds [13]. Exogenous enzyme supplementation also lowers the problem of wet digesta and reduces the wet litter problem in the poultry birds fed with high amount of NSPs in the feed [14]. However, previous studies reported that storage reduces the antinutritional factors especially NSPs and phytate in plant-based feed ingredient and inclusion of stored plant-based feed ingredients enhance the performance of the birds [15,16,17,18,19,20]. Therefore, studies to investigate the impact of exogenous enzymes, especially NSPase and phytase on performance of birds fed stored wheat-based diet is mandatory. Although, few studies have investigated the impact of supplementation of exogenous enzymes especially non structural polysaccharidases (NSPase) and phytase on performance of birds fed stored wheat-based diet, however, according to our knowledge, none of the study has been conducted to evaluate the impact of supplementation of exogenous enzymes in the diet contained 1.5 years and 2.5 years old wheat. This study is of particular interest in the countries where wheat is a staple food, and it is stored to maintain food supply chain especially during scarcity period like in Pakistan; and in this respect the Government of Pakistan stored huge amount of wheat every year. After a couple of years, surplus stored wheat is provided to poultry feed manufacturer at lower price and poultry feed manufacturers use stored wheat in diet of poultry. We hypothesized that storage of wheat for 1.5 and 2.5 years will not require exogenous enzyme in the diet of broilers. Therefore, the objective of the present study was to evaluate the effects of stored wheat-based diet (1.5 and 2.5 years stored wheat) with and without the supplementation of xylanase and phytase individually or in combination on the growth performance, digestibility, digesta viscosity and carcass characteristics of broiler chickens.

## 2. Materials and Methods

Experimental protocols were approved by Synopsis Committee and Office of Director Graduate Studies, Agriculture University, Faisalabad, Pakistan (939#12-10-2021).

### 2.1. Birds Management and Experimental Treatments

One month before the arrival of chicks, the shed was cleaned thoroughly and fumigated. A total of 640-day-old male broiler chicks (Ross-308) were procured from Arslan Chicks (Pvt Ltd. 48-C, Satellite Town, Murree Road, Near Chandni Chowk, Rawalpindi-Pakistan). A total of eight dietary treatments were used in this experiment. Experimental diets were formulated as, 1.5 years old wheat-based diets with (0 Xylanase 0 Phytase), (1 Xylanase 0 Phytase), (0 Xylanase 1 Phytase), and (1 Xylanase 1 phytase) and 2.5 years old wheat based diets with (0 Xylanase 0 Phytase), (1 Xylanase 0 Phytase), (0 Xylanase 1 Phytase), and (1 Xylanase 1 phytase) supplementation.

The composition of ingredient and their related data used in the formulation of experimental diets were taken from Brazilian Tables for Poultry and Swine [21]. PlosXylan^®^ enzymes were supplemented in the experimental rations 1 Xyln 0 Phyt and 1 Xyln 1 phyt (1.5 years old and 2.5 years old wheat) @ 50 kg/ton following the instruction of the supplier. The exogenous enzymes (GenPhy^®^) were supplemented in the experimental ration 1 Xyln 0 Phyt, 1 Xyln 1 phyt (1.5 years old and 2.5 years old wheat) @ 100 kg/ton following the instruction of the supplier. PlosXylan^®^ and GenPhy^®^ are enzymes product of a Chinese Company (Shandong Enzymes Technologies, Linyi, China), obtained from Polaris Life Sciences (Pvt. Ltd.) 182, Shadman-II Lahore Punjab, Pakistan.

Ingredients inclusion level and chemical composition of starter and finisher diets are presented in Table 1 and Table 2. Each of the eight dietary treatments was randomly assigned to the 64 pens, each pen had 10 birds. Pens were covered with three-inch wood shavings for chicks bedding. During the experiment, chicks was reared for 35 days keeping the same environmental conditions for all treatments. Fresh and clean water was offered round the clock. Birds was vaccinated according to local vaccination program. A circular bottom feeder was provided for each pen, and nipple drinking system allowed for continuous water availability.

### 2.2. Data Collection for Growth Performance

Birds were individually weighed and assigned to 64 pens (10 birds per pen). On a weekly, basis body weight was measured to record for body weight gain. Estimation of feed intake was done by subtracting the amount of feed refused from the total feed offered during the course of the week. Feed intake and weight was used to calculate feed conversion ratio (FCR), as described in the study of [22].

### 2.3. Digestibility Assay

A digestibility assay was carried out. Experimental birds were fed diets mixed with Acid insoluble ash (Celite^®^, Shanghai, China). Diets were analyzed for dry matter (DM), crude protein (CP), ether extract (EE), crude fiber, and ash [23]. This feeding plan continued till the end of the trial. Fecal digesta samples were collected for digestibility assay on day 35. The floor of each pen was covered with plastic sheets on day 34 to prevent contamination from bedding material. Excreta was completely cleaned of feathers and other extraneous objects from each pen. Each pen’s excreta were weighed, homogenized, dried in the oven, and ground for further chemical analysis. Samples were dried in hot air oven at 105 °C for 2 h for dry matter determination. Ash was estimated by burning each sample in muffle furnace at temperature 650 °C for 2 h. Ether was used for determination of ether extract. Crude protein was calculated by multiplying nitrogen (through Kjeldahl method) with 6.25 [23]. Nutrient digestibility was determined using following formula
$$\mathrm{Digestibility}\;(\%)=100\;-\;(100\;\times \;\frac{\%\;\mathrm{marker}\;\mathrm{in}\;\mathrm{feed}}{\%\;\mathrm{marker}\;\mathrm{in}\;\mathrm{feces}}\times \frac{\%\;\mathrm{nutrient}\;\mathrm{in}\;\mathrm{feces}}{\%\;\mathrm{nutrient}\;\mathrm{in}\;\mathrm{feed}})$$

### 2.4. Data Collection for Carcass Characteristics

At the end of trial, two birds from each treatment were selected at random and weighed individually and slaughtered. In order to get data on carcass characteristics, lthe ive body weight of birds was recorded. After slaughtering, feathers were removed followed by evisceration.

Dressing percentage

The carcass was weighed and recorded live weight data was used to calculate dressing percentage by using the following formula.
$$\mathrm{Dressing}\;(\%)=\frac{\mathrm{Carcass}\;\mathrm{weight}\;(g)}{\mathrm{Live}\;\mathrm{weight}\;(g)}\times 100$$

Breast weight

Slaughtered birds’ breasts were separated and weighed to obtain breast weight.
$$\mathrm{Breast}\;(\%)=\frac{\mathrm{Breast}\;\mathrm{weight}\;(g)}{\mathrm{Carcass}\;\mathrm{weight}\;(g)}\times 100$$

Thigh weight

Slaughtered birds’ thighs were separated and weighed to obtain thigh weight.
$$\mathrm{Thigh}\;(\%)=\frac{\mathrm{Thigh}\;\mathrm{weight}\;(g)}{\mathrm{Carcass}\;\mathrm{weight}\;(g)}\times 100$$

### 2.5. Determination of Viscosity

Two broiler birds were slaughtered for the purpose of collecting digesta in order to determine the viscosity of samples. The digesta contents of the gastrointestinal tract (except for the proventriculus) was collected gently by finger, stripping each GI tract segment and subsequently frozen at −20 °C for future analysis. Digesta samples were pooled in a falcon tube per replicate. The viscosity of digesta samples were determined by using a Brookfield DV-E viscometer (Brookfield Engg., Middleboro, MA, USA). The tubes were centrifuged for 5 min at 3000 rpm and then the supernatant was centrifuged for 5 min at 12,500 rpm. The viscometer was preheated at 25 °C then digesta supernatant was introduced in the viscometer. The average shear rate used for measuring viscosity ranged between 45.0 s^−1^ and 22.5 s^−1^ [12,24].

### 2.6. Statistical Analysis

Data collected were analyzed using General Linear Model procedure under 2 × 2 × 2 factorial arrangement in a completely randomized design [25]. The model statement was:Y = μ + Wi + Xj + Pk + (W × X)ij + (W × *P*)ik + (X × *P*)jk + (W × X × K)ijk + εijk
where

Y = Any of dependent variable tested in study

μ = Overall mean

Wi = Wheat stored for either 1.5 or 2.5 years

Xj = Level of Xylanase either 0 or 1

Pk = Level of phytase either 0 or 1

(W × X)ij = Interaction between wheat storage time and Xylanase level

(W × *P*)ik = Interaction between wheat storage time and Phytase level

(X × *P*)jk = Interaction between Xylanase and Phytase level

(W × X × *P*)ijk = Interaction between wheat storage time, Xylanase and Phytase levels

Εijk = Residual error

All main effects and interactions were declared significant at *p* < 0.05.

## 3. Results

### 3.1. Growth Performance

Growth performance data of supplementation of exogenous enzymes xylanase and phytase alone or in combination in stored wheat-based diet is presented in Table 3. In starter phase (1–21 days), and over all phase (1–35 days) diets with the supplementation of xylanase and phytase have significant effect on feed intake (*p* < 0.05). Diets supplemented with xylanase and phytase in combination had higher intake of feed as compared to the experimental diets that were without supplementation of xylanase and phytase (*p* < 0.05). In starter phase 1.5 years old wheat with 0 xyln 0 phyt, reduced the body weight gain (*p* < 0.05), while 1.5 years old wheat with 1 xyln 1 phyt, had significantly increased body weight gain as compared to the other dietary treatments (*p* < 0.05). 21st day FCR showed that an interaction was found between wheat and phytase. In overall phase (1–35 days), experimental diet contained 1.5 years old and 2.5 years old-wheat with exogenous enzymes supplementation individually or in combination increased the body weight gain (*p* < 0.05). 1.5 years old wheat-based diet supplemented with xylanase and phytase had increased weight gain as compared to the other 1.5 years old wheat-based diets with or without xylanase and phytase (*p* < 0.05). In the starter phase, diets with and without the supplementation of xylanase and phytase individually or in combination have no effect on FCR (*p* > 0.05). In finisher phase (22–35 days) supplementation of xylanase and phytase individually or in combination in 1.5 years old wheat-based diet and 2.5 years old-wheat based diets significantly improve the FCR (*p* < 0.05). While 1.5 years old and 2.5 years old wheat-based diets without the supplementation of xylanase and phytase individually or in combination had poor FCR (*p* < 0.05) as compared to the other dietary treatments in finisher phase (22–35 days). 35th day FCR showed that an interaction was found between xylanase and phytase.

### 3.2. Nutrient Digestibility

Data of stored wheat-based diets with and without the supplementation of xylanase and phytase for nutrient digestibility is presented in Table 4. Dry matter digestibility results explored that 1.5 years old wheat-based diet and 2.5 years old wheat-based diet without the supplementation of xylanase and phytase had significantly lowered dry matter digestibility (*p* < 0.05). Supplementation of xylanase and phytase to 1.5 years old wheat-based diet and 2.5 years old wheat-based diets significantly enhanced the dry matter digestibility (*p* < 0.05). Data on CP, CF, EE and ash digestibility explored that supplementation of xylanase and phytase in combination to 1.5 years old wheat-based diet and 2.5 years old wheat-based diet significantly enhanced the digestibility (*p* < 0.05), these diets without the supplementation of xylanase and phytase had significantly lowered CP, CF, EE and ash digestibility (*p* < 0.05). DM digestibility results showed that an interaction was found between the xylanase and phytase whereas ash digestibility results showed an interaction between wheat and xylanase.

### 3.3. Viscosity and Carcass Parameters

The data of digesta viscosity of the current experiment is presented in Table 5. Results explored that 1.5 years old wheat-based diet and 2.5 years old wheat-based diet without the supplementation of xylanase and phytase had significantly higher digesta viscosity (*p* < 0.05). Supplementation of xylanase and phytase in 2.5 years old wheat-based diet and 1.5 years old wheat-based diet significantly lowered the digesta viscosity (*p* < 0.05). The results of carcass parameters are presented in Table 6. Results of carcass were not influenced by experimental treatments.

## 4. Discussion

New season grains are problematic for broiler production due to the high contents of soluble NSPs that are responsible for increasing the digesta viscosity [16]. However, storing grains for three to four months improved nutritive value and has a positive impact on broiler chicken production [16,17]. Improving nutritional value of stored grains is due to the activation of their endogenous enzymes especially phytase and NSPs degrading enzymes in barley and wheat. The inclusion of stored wheat with supplementation of exogenous enzymes in the diet of poultry further reduced negative issues of anti-nutritional factors and enhanced the performance of poultry birds. In recent years, scientists took huge attention to explore the possible interactions between exogenous phytase and xylanase enzymes in wheat-based broiler feeds [3,4].

In the present study, the inclusion of both xylanase and phytase enzymes in combination had positive impact on intake and growth performance in terms of body weight gain of broiler birds in the starter phase as well as in the overall phase.

On day 21, greater weight gain was observed both in birds fed with 1.5-year-old wheat-based diet contained both xylanase and phytase enzymes in combination and 2.5-year-old wheat-based diet. Phytase was expected to increase nutrient utilization and performance because: (i) its substrate phytate contains around 2/3 of the total phosphorus in ingredients obtained from plant sources [18], which is least digestible by broilers [19], (ii) Phytate in its natural state is complexed with minerals and proteins, and thus it can reduce the availability of these nutrients for utilization [20,26]; and (iii) Phytate is negatively charged at all pH conditions in the GIT and thus can bind or have ability to bind nutrients with positive charge and some enzymes (endogenous) in the GIT, thereby having strong potential to reduce nutrient utilization in the body [27,28]. Xylanase was also expected to increase the nutrient utilization and performance because: (i) its substrate (arabinoxylans) reduces nutrient intake and digestibility by increasing digesta viscosity and decreasing passage rate; and (ii) in wheat, arabinoxylans are the major component of NSPs, which reduces nutrient availability by encapsulation [29,30,31]. In the current study, supplementation of exogenous enzymes phytase and xylanase in combination was expected to further increase nutrient utilization and performance because it has been reported that phytate contents in wheat is extremely concentrated in aleurone cells [32] and cell walls of aleurone cells are known to made of chiefly arabinoxylans [33], and thus it could be speculated that xylanase enzymes in the current study hydrolyzed arabinoxylans to break the cell walls and increased approaches of exogenous phytase to phytate [34] and ultimately it was responsible for better nutrient utilization in the current study. These results are also in accordance with the study of Selle et al. [35], they reported that supplementation of xylanase in combination with phytase significantly enhanced feed efficiency. The results of feed conversion ratio also explored those birds fed with diet without the supplementation of xylanase and phytase in 1.5 years old and 2.5 years old wheat based diet had poor feed conversion ratio because of the fact that there was no supplementation of xylanase and phytase for the destruction of NSPs and for the availability of bounded minerals that may result in high digesta viscosity and lead to the impaired utilization of nutrients [20,36].

At day 35, greater weight gain and FCR were observed in the birds fed with both a 1.5 years old and a 2.5 years old wheat-based diets supplemented both with xylanase and phytase. Better performance by supplementation of xylanase and phytase in stored wheat-based diet could be explained by the action of xylanase, that reduced digesta viscosity and released nutrients entrapped within the cell wall matrix, that could increase the access of phytase to its substrate and facilitate the absorption of liberated nutrients [20]. Similar to our study, Ravindran et al. [37] found that individual additions of phytase and xylanase improved apparent metabolizable energy AME of wheat by 9.7 and 5.3%, respectively. When the diet was supplemented with a combination of the two enzymes, the apparent metabolizable energy was improved by 19.0%. The improvements in weight gain and feed efficiency in birds given wheat-based diets with individual additions of xylanase or phytase go parallel with reductions in the viscosity of digesta. These results are in agreement with other reports [38,39,40]. According to Wu et al. [41], supplementary phytase increased weight and feed efficiency by 17.5 and 29%, respectively. It is generally hypothesized that NSPs have detrimental effects on viscosity and bird performance, and that exogenous xylanase and phytase enzymes can help to mitigate these effects [29,42,43] and enhance the broiler performance.

The boost in the performance of broiler birds by supplementation of exogenous enzymes could also be explained by complete decomposing of the NSPs after partial cleavage of bonds in storage in monosaccharides and a subsequent absorption of the released sugars by the animal. It might be assumed that polymer’s partial cleavage happened in wheat during storage, which removes their anti-nutritive properties. Therefore, exogenous enzymes were enough to split the NSPs and result in a significant improvement in nutrient digestion and absorption in the gut that at the end lead to the better performance of broiler birds [44,45].

Less viscosity was observed in the birds fed with the 100% − 2.5 YOW + Xyl + Phy diet and the 100% − 352 1.5 YOW + Xyl + Phy one (*p* < 0.05). High viscosity was observed in birds fed with diets without the supplementation of xylanase and phytase in 1.5 and 2.5 years old wheat diets (*p* < 0.05). Results of our study are in accordance with the study of Engberg et al. [21]; they reported that supplementation of xylanase in wheat-based diets, reduced the viscosity of illeal chyme. Similar results of digesta viscosity reported by Adeola and Bedford [46], they observed improved performance of White Pekin ducks after xylanase supplementation to wheat-based diets when the wheat was highly viscous (45.68 cps), but not when it was less viscous (5.86 cps). It is known that higher viscosity might promote the development of anaerobic bacteria. According to Choct et al. [47], xylanase supplementation to wheat-based diets lowered the microbial activity of ileal digesta as indicated by lower amounts of volatile fatty acids. According to Sinlae and Choct [48], the presence of unwanted organisms such Clostridium perfringens in caecal contents was decreased when xylanase was added to a diet based on wheat. Supplementation of xylanase in high ileal viscosity is reported to be more beneficial while had low effect in medium or low viscosity [49]. When the digesta viscosity is less than 10 mPa.s. it becomes a less significant factor in predicting the performance response of birds given wheat-based diets as highlighted by Bedford et al. [29].

It is interesting to note that addition of phytase and xylanase in 2.5 years old wheat based greatly lowered the viscosity of the digesta in ileum. Two explanations may be proposed to clarify these outcomes. First, the xylanase product used in the current study may have good xylanase activity and degraded the xylan content of diet. Second, it is expected that microbial phytase damages the wheat cell wall matrix similarly to exogenous xylanase [37], degrading the NSPs and lowering the viscosity of digesta in broiler birds fed high fiber wheat-based diets. Sinlae and Choct [48] also reported that addition of xylanase and phytase to the basal diet contributed to a decrease in the viscosity of broiler birds digesta from the duodenum. Similarly, other studies have reported that the combination of xylanase and phytase results in reductions in viscosity of digesta [50,51,52,53] and enhance the performance of the birds.

In the current study, better dry matter digestibility was observed in birds fed with 1.5 years old wheat and 2.5 years old wheat supplemented with xylanase and phytase; similarly, high dry matter digestibility in broiler chickens fed with barley, oats and wheat with exogenous enzymes has also been reported by the researchers [54,55]. Similarly with the findings of dry matter digestibility, CP digestibility was better in birds fed with 2.5 years old wheat with 1 xyln 1 phyt and 1.5 years old wheat with 1 xyln 1 phyt diets. Woyengo et al. documented that xylanase improved apparent amino acid digestibility [56], which suggest that wheat NSPs impair the digestibility of amino acid [57]. It has been reported that arabinoxylans encapsulate amino acids in the grain preventing amino acids from digestion and absorption in the small intestine or promoting the increase of endogenous amino acids [53]. Therefore, supplementing xylanase in the diet may degrade arabinoxylans complex in wheat and enhanced protein digestibility in the current study. According to Ravindran, Selle and Bryden [37], the simultaneous addition of phytase and xylanase to wheat-based broiler diets was advantageous due to wheat’s increased apparent metabolizable energy and improved protein digestibility. Advantage of xylanase and phytase in combination has been previous reported by researcher and they reported that phytase and xylanase in combination to phosphorus deficient broiler diets, considerably enhanced growth rates, but when added separately, they had little impact [58]. Therefore, in the current study, the combination of the two enzymes worked much better to considerably enhance the growth performance that may be due to enhanced AME. In the present study, combined effects of phytase and xylanase on crude fiber digestibility were significant and enhanced crude fiber digestibility could be the result of degradation of the hemicellulose of crude fiber by exogenous xylanase, as reported by Pettersson and Aman [59].

Ether extract digestibility was also improved significantly with the supplementation of enzymes in stored wheat diets (1.5 years old and 2.5 years old wheat). It seems that xylanase breaks the cell wall and liberates more nutrients for digestion and absorption in the gastrointestinal tract of the poultry birds [60]. Svihus and Gullord [61] and Meng and Slominski [62] reported that improvement in the absorption of energy-producing nutrients, i.e., fat, protein, and principally starch may provide more energy for maintenance and growth. Ash represents mainly the mineral contents of diet and higher digestibility was observed in diet supplemented xylanase and phytase in 2.5 years old wheat diet that could be due to the improvements in nutritional value by the supplementation of exogenous enzymes [62].

## 5. Conclusions

Based on the findings of the current study, it is concluded that supplementation of wheat stored for 1.5 and 2.5 years with xylanase and phytase in combination had pronounced improvements in the growth performance of broiler birds due to the synergistic effect of xylanase and phytase with diet.

## Figures and Tables

**Table 1 animals-13-00278-t001:** Ingredients composition of experimental diets (% inclusion level) for starter phase.

Ingredients%	1.5 Years Old Wheat				2.5 Years Old Wheat			
	0 Xyln 0 Phyt	1 Xyln 0 Phyt	0 Xyln 1 Phyt	1 Xyln 1 phyt	0 Xyln 0 Phyt	1 Xyln 0 Phyt	0 Xyln 1 Phyt	1 Xyln 1 phyt
Wheat	57.52	57.52	57.52	57.52	57.52	57.52	57.52	57.52
Soybean Meal 46%	29.89	29.89	29.89	29.89	29.89	29.89	29.89	29.89
Canola Meal	3.49	3.49	3.49	3.49	3.49	3.49	3.49	3.49
Poultry By-product Meal	3.20	3.20	3.20	3.20	3.20	3.20	3.20	3.20
Limestone	1.00	1.00	1.00	1.00	1.00	1.00	1.00	1.00
Rice Polish	3.57	3.57	3.57	3.57	3.57	3.57	3.57	3.57
Monocalcium Phospahte	0.34	0.34	0.34	0.34	0.34	0.34	0.34	0.34
Lysine Sulfate 70%	0.05	0.05	0.05	0.05	0.05	0.05	0.05	0.05
Methionine 99%	0.21	0.21	0.21	0.21	0.21	0.21	0.21	0.21
Sodium Chloride	0.17	0.17	0.17	0.17	0.17	0.17	0.17	0.17
Sodium Bicarbonate	0.09	0.09	0.09	0.09	0.09	0.09	0.09	0.09
Premix	0.44	0.44	0.44	0.44	0.44	0.44	0.44	0.44
L-Threonine 98%	0.03	0.03	0.03	0.03	0.03	0.03	0.03	0.03
Total	100	100	100	100	100	100	100	100
	Nutrient composition (g/kg) of experimental diets for 1–21 days							
Crude Protein	23.07	23.07	23.07	23.07	23.07	23.07	23.07	23.07
Metabolizable energy (Kcal/kg)	2987	2987	2987	2987	2987	2987	2987	2987
Calcium	0.96	0.96	0.96	0.96	0.96	0.96	0.96	0.96
Available Phosphorus	0.48	0.48	0.48	0.48	0.48	0.48	0.48	0.48
Sodium	0.18	0.18	0.18	0.18	0.18	0.18	0.18	0.18
Chloride	0.22	0.22	0.22	0.22	0.22	0.22	0.22	0.22
Methionine (D)	0.51	0.51	0.51	0.51	0.51	0.51	0.51	0.51
Methionine plus cysteine	0.95	0.95	0.95	0.95	0.95	0.95	0.95	0.95
Lysine	1.22	1.22	1.22	1.22	1.22	1.22	1.22	1.22
L-Threonine	0.86	0.86	0.86	0.86	0.86	0.86	0.86	0.86
Tryptophan	0.20	0.20	0.20	0.20	0.20	0.20	0.20	0.20
Arginine	1.38	1.38	1.38	1.38	1.38	1.38	1.38	1.38
L-Valine	0.96	0.96	0.96	0.96	0.96	0.96	0.96	0.96

0 Xyln 0 Phyt = Xylanase (0 XU), Phytase (0 FTU); 1 Xyln 0 Phyt = Xylanase (500 XU), Phytase (0 FTU); 0 Xyln 1 Phyt = Xylanase (0 XU), Phytase (500 FTU); 1 Xyln 1 phyt = Xylanase (500 XU) + Phytase (500 FTU); The exogenous enzymes (PlosXylan^®^) was supplemented in the experimental rations 1 Xyln 0 Phyt and 1 Xyln 1 phyt (1.5 years old and 2.5 years old wheat) @ 50 kg/ton following the instruction of the supplier. The exogenous enzymes (GenPhy^®^) were supplemented in the experimental ration 1 Xyln 0 Phyt, 1 Xyln 1 phyt (1.5 years old and 2.5 years old wheat) @ 100 kg/ton following the instruction of the supplier. PlosXylan^®^ and GenPhy^®^ are enzymes product of Chinese Company (Shandong Enzymes Technologies, Linyi, China), obtained from Polaris Life Sciences (Pvt. Ltd.), Pakistan.

**Table 2 animals-13-00278-t002:** Ingredients composition of experimental diets (% inclusion level) for finisher phase.

Ingredients%	1.5 Years Old Wheat				2.5 Years Old Wheat			
	0 Xyln 0 Phyt	1 Xyln 0 Phyt	0 Xyln 1 Phyt	1 Xyln 1 phyt	0 Xyln 0 Phyt	1 Xyln 0 Phyt	0 Xyln 1 Phyt	1 Xyln 1 phyt
Wheat	62.04	62.04	62.04	62.04	62.04	62.04	62.04	62.04
Soybean Meal 46%	23.24	23.24	23.24	23.24	23.24	23.24	23.24	23.24
Poultry By-product Meal	1.55	1.55	1.55	1.55	1.55	1.55	1.55	1.55
Limestone	0.76	0.76	0.76	0.76	0.76	0.76	0.76	0.76
Rice Polish	8.07	8.07	8.07	8.07	8.07	8.07	8.07	8.07
Poultry Oil	2.82	2.82	2.82	2.82	2.82	2.82	2.82	2.82
Monocalcium Phospahte	0.19	0.19	0.19	0.19	0.19	0.19	0.19	0.19
Lysine Sulfate 70%	0.34	0.34	0.34	0.34	0.34	0.34	0.34	0.34
Methionine 99%	0.23	0.23	0.23	0.23	0.23	0.23	0.23	0.23
Sodium Chloride	0.17	0.17	0.17	0.17	0.17	0.17	0.17	0.17
Sodium Bicarbonate	0.09	0.09	0.09	0.09	0.09	0.09	0.09	0.09
Premix	0.44	0.44	0.44	0.44	0.44	0.44	0.44	0.44
L-Threonine 98%	0.06	0.06	0.06	0.06	0.06	0.06	0.06	0.06
Total	100	100	100	100	100	100	100	100
	Nutrient composition (g/kg) of experimental diets for 22–35 days							
Crude Protein	18.40	18.40	18.40	18.40	18.90	18.90	18.90	18.90
Metabolizable energy (Kcal/kg)	3205	3205	3205	3205	3197	3197	3197	3197
Calcium	0.79	0.79	0.79	0.79	0.79	0.79	0.79	0.79
Available. Phosphorus	0.39	0.39	0.39	0.39	0.39	0.39	0.39	0.39
Sodium	0.20	0.20	0.20	0.20	0.20	0.20	0.20	0.20
Chloride	0.70	0.70	0.70	0.70	0.70	0.70	0.70	0.70
Methionine (D)	0.43	0.43	0.43	0.43	0.43	0.43	0.43	0.43
Methionine plus cystine	0.80	0.80	0.80	0.80	0.80	0.80	0.80	0.80
Lysine	1.03	1.03	1.03	1.03	1.03	1.03	1.03	1.03
L-Threonine	0.69	0.69	0.69	0.69	0.69	0.69	0.69	0.69
Tryptophan	0.16	0.16	0.16	0.16	0.16	0.16	0.16	0.16
Arginine	1.10	1.10	1.10	1.10	1.10	1.10	1.10	1.10
Isoleucine	0.71	0.71	0.71	0.71	0.71	0.71	0.71	0.71
L-Valine	0.78	0.78	0.78	0.78	0.78	0.78	0.78	0.78

0 Xyln 0 Phyt = Xylanase (0 XU), Phytase (0 FTU); 1 Xyln 0 Phyt = Xylanase (500 XU), Phytase (0 FTU); 0 Xyln 1 Phyt = Xylanase (0 XU), Phytase (500 FTU); 1 Xyln 1 phyt = Xylanase (500 XU) + Phytase (500 FTU); The exogenous enzymes (PlosXylan^®^) was supplemented in the experimental rations 1 Xyln 0 Phyt and 1 Xyln 1 phyt (1.5 years old and 2.5 years old wheat) @ 50 kg/ton following the instruction of the supplier. The exogenous enzymes (GenPhy^®^) were supplemented in the experimental ration 1 Xyln 0 Phyt, 1 Xyln 1 phyt (1.5 years old and 2.5 years old wheat) @ 100 kg/ton following the instruction of the supplier. PlosXylan^®^ and GenPhy^®^ are enzymes product of Chinese Company (Shandong Enzymes Technologies, Linyi, China), obtained from Polaris Life Sciences (Pvt. Ltd.), Pakistan.

**Table 3 animals-13-00278-t003:** Growth performance of broiler birds fed with stored wheat-based diet with and without the supplementation of xylanase and phytase.

Items	1.5 Years Old Wheat				2.5 Years Old Wheat				SEM	*p*-Values						
	0 Xyln 0 Phyt	1 Xyln 0 Phyt	0 Xyln 1 Phyt	1 Xyln 1 phyt	0 Xyln 0 Phyt	1 Xyln 0 Phyt	0 Xyln 1 Phyt	1 Xyln 1 Phyt		Wheat	Xylanase	Phytase	Wheat × Xylanase	Wheat × Phytase	Xylanase × Phytase	Wheat × Xylanase × Phytase
0–21 day Feed intake (g)	1255.1	1274.8	1268.5	1284.8	1254.1	1272.3	1264.0	1287.3	8.70	0.753	0.031	0.037	0.753	0.931	0.920	0.627
Weight gain (g)	895.36	908.40	910.86	933.85	910.47	921.02	903.81	921.71	8.50	0.7531	0.027	0.045	0.7531	0.9316	0.920	0.879
Feed conversion ratio	1.41	1.40	1.39	1.37	1.37	1.38	1.40	1.40	0.004	0.544	0.685	0.981	0.587	0.027	0.477	0.697
22–35 day Feed intake (g)	2160.3	2181.1	2176.6	2199.0	2160.1	2179.3	2172.8	2195.1	4.13	0.243	0.019	0.041	0.833	0.489	0.567	0.833
Weight gain (g)	1216.5	1251.5	1246.4	1271.0	1216.4	1248.7	1243.0	1268.3	3.99	0.2430	0.001	0.015	0.8330	0.4894	0.5677	0.685
Feed conversion ratio	1.77	1.74	1.74	1.73	1.77	1.75	1.75	1.73	0.007	0.372	0.011	0.021	0.801	0.994	0.001	0.546

0 Xyln 0 Phyt = Xylanase (0 XU), Phytase (0 FTU); 1 Xyln 0 Phyt = Xylanase (500 XU), Phytase (0 FTU); 0 Xyln 1 Phyt = Xylanase (0 XU), Phytase (500 FTU); 1 Xyln 1 phyt = Xylanase (500 XU) + Phytase (500 FTU); SME: Standard Error of Mean.

**Table 4 animals-13-00278-t004:** Nutrient digestibility of broiler birds fed with stored wheat-based diet with and without the supplementation of xylanase and phytase.

Items	1.5 Years Old Wheat				2.5 Years Old Wheat				SEM	*p*-Values						
	0 Xyln 0 Phyt	1 Xyln 0 Phyt	0 Xyln 1 Phyt	1 Xyln 1 phyt	0 Xyln 0 Phyt	1 Xyln 0 Phyt	0 Xyln 1 Phyt	1 Xyln 1 Phyt		Wheat	Xylanase	Phytase	Wheat × Xylanase	Wheat× Phytase	Xylanase × Phytase	Wheat × Xylanase × Phytase
Dry matter digestibility	61.15	65.57	66.46	67.39	60.87	65.97	67.36	68.070	1.21	0.501	0.001	0.013	0.854	0.560	0.011	0.720
Crude protein digestibility	64.71	71.50	71.63	75.60	65.99	71.43	73.610	77.325	1.19	0.074	0.021	0.033	0.520	0.326	0.092	0.655
Crude fiber digestibility	62.265	71.525	69.850	77.510	62.09	71.59	70.14	77.18	1.37	0.963	0.044	0.049	0.893	0.980	0.177	0.761
Ether extract digestibility	77.300	81.525	82.71	87.33	75.26	82.06	83.48	87.50	1.29	0.826	0.011	0.017	0.449	0.353	0.364	0.236
Ash digestibility	60.335	67.360	71.285	74.180	63.730	71.600	68.170	75.810	1.36	0.054	0.020	0.031	0.075	0.010	0.149	0.191

DM = dry matter, CP = crude protein, CF = crude fiber, EE = ether extract; 0 Xyln 0 Phyt = Xylanase (0 XU), Phytase (0 FTU); 1 Xyln 0 Phyt = Xylanase (500 XU), Phytase (0 FTU); 0 Xyln 1 Phyt = Xylanase (0 XU), Phytase (500 FTU); 1 Xyln 1 phyt = Xylanase (500 XU) + Phytase (500 FTU); SME: Standard Error of Mean

**Table 5 animals-13-00278-t005:** Effect of stored wheat supplementation with and without xylanase and phytase on digesta viscosity of broiler birds at day 35.

Items	1.5 Years Old Wheat				2.5 Years Old Wheat				SEM	*p*-Values						
	0 Xyln 0 Phyt	1 Xyln 0 Phyt	0 Xyln 1 Phyt	1 Xyln 1 phyt	0 Xyln 0 Phyt	1 Xyln 0 Phyt	0 Xyln 1 Phyt	1 Xyln 1 Phyt		Wheat	Xylanase	Phytase	Wheat × Xylanase	Wheat × Phytase	Xylanase × Phytase	Wheat × Xylanase × Phytase
Viscosity (cps)	15.61	9.77	9.71	6.68	14.25	9.09	9.19	6.30	1.72	0.962	0.031	0.039	0.675	0.891	0.049	0.654

0 Xyln 0 Phyt = Xylanase (0 XU), Phytase (0 FTU); 1 Xyln 0 Phyt = Xylanase (500 XU), Phytase (0 FTU); 0 Xyln 1 Phyt = Xylanase (0 XU), Phytase (500 FTU); 1 Xyln 1 phyt = Xylanase (500 XU) + Phytase (500 FTU); SME: Standard Error of Mean.

**Table 6 animals-13-00278-t006:** Effect of stored wheat supplementation with and without xylanase and phytase on carcass characteristics of broiler birds at day 35.

Carcass Characteristics	1.5 Years Old Wheat				2.5 Years Old Wheat				SEM	*p*-Values						
	0 Xyln 0 Phyt	1 Xyln 0 Phyt	0 Xyln 1 Phyt	1 Xyln 1 phyt	0 Xyln 0 Phyt	1 Xyln 0 Phyt	0 Xyln 1 Phyt	1 Xyln 1 Phyt		Wheat	Xylanase	Phytase	Wheat × Xylanase	Wheat × Phytase	Xylanase × Phytase	Wheat × Xylanase × Phytase
Dressing%	53	57	55	60	51	55	57	61	5.69	0.501	0.052	0.131	0.854	0.560	0.110	0.720
Breast%	20	21	22	25	21	22	23	25	3.71	0.071	0.087	0.081	0.769	0.661	0.442	0.320
Thigh%	9	10	10	12	11	10	11	13	2.16	0.063	0.077	0.213	0.601	0.061	0.052	0.205
Liver%	2.0	2.1	2.1	2.4	2.1	2.2	2.3	2.4	0.03	0.061	0.152	0.231	0.844	0.601	0.190	0.420
Heart%	0.42	0.44	0.43	0.50	0.44	0.48	0.46	0.56	0.07	0.812	0.770	0.513	0.051	0.081	0.073	0.201
Gizzard%	1.20	1.21	1.22	1.26	1.21	1.22	1.24	1.30	0.16	0.061	0.067	0.342	0.569	0.061	0.053	0.320

0 Xyln 0 Phyt = Xylanase (0 XU), Phytase (0 FTU); 1 Xyln 0 Phyt = Xylanase (500 XU), Phytase (0 FTU); 0 Xyln 1 Phyt = Xylanase (0 XU), Phytase (500 FTU); 1 Xyln 1 phyt = Xylanase (500 XU) + Phytase (500 FTU); SME: Standard Error of Mean.

## Data Availability

Data will be available by corresponding author.

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
