# Peer review of "Impact of Exogenous Xylanase and Phytase, Individually or in Combination, on Performance, Digesta Viscosity and Carcass Characteristics in Broiler Birds Fed Wheat-Based Diets"

_animals, 2023, doi:10.3390/ani13020278_

Round 1

Reviewer 1 Report

The manuscript entitled (Impact of exogenous xylanase and phytase, individually or in 2 combination, on performance, digesta viscosity and carcass characteristics in broiler birds fed wheat based diets) is well written and contains valuable data, however some points should be considered before publication.

I think the data should be analysed with GLM with factorial design instead of CRD, this point very critical, here you should estimate the independent and interaction effects and their significance and the used model should be written.

The authors used stored wheat based diets instead of corn as a source of energy, where is the economic values in your data, it is very important.

Table 3 lacks the data of body weight, only contains weight gain. the initial and final body weight are very important.

Table 6, what did you mean by 0-35 in carcass traits. (is that true??)

Author Response

The manuscript entitled (Impact of exogenous xylanase and phytase, individually or in 2 combination, on performance, digesta viscosity and carcass characteristics in broiler birds fed wheat based diets) is well written and contains valuable data, however some points should be considered before publication.

Dear Reviewer,

Thanks a lot for your positive comment and compliment on our PhD research work. I really appreciate your effort and time to review our manuscript. I know, your suggestions are valuable and for the sake of improvement of the manuscript. Therefore, I have tired my best to follow your comments and revise the manuscript. I hope you will find manuscript suitable for publication this time.

I think the data should be analyzed with GLM with factorial design instead of CRD, this point very critical, here you should estimate the independent and interaction effects and their significance and the used model should be written.

Ans: we have reanalyzed the data as suggested by you. You can find it in the revised manuscript in statistical analysis, results tables, results sections and somewhere in the discussion.

Now statistical analysis statement seems as given below in the revised manuscript at line 195-211

Data collected were analyzed using General Linear Model procedure under 2*2*2 factorial arrangement in a completely randomized design. The model statement was:

Y=μ+Wi+Xj+Pk+(W*X)ij+(W*P)ik+ (X*P)jk+(W*X*K)ijk+ εijk

Where,

Y= Any of dependant variable tested in study

μ=Overall mean

Wi=Wheat stored for either 1.5 or 2.5 years

Xj=Level of Xylanase either 0 or  1

Pk= Level of phytase either 0 or 1

(W*X)ij= Interaction between wheat storage time and Xylanase level

(W*P)ik= Interaction between wheat storage time and Phytase level

(X*P)jk= Interaction between Xylanase and Phytase level

(W*X*P)ijk=Interaction between wheat storage time, Xylanase and Phytase levels

εijk= Residual error

All main effects and interactions were declared significant at P<0.05

The authors used stored wheat based diets instead of corn as a source of energy, where is the economic values in your data, it is very important.

Ans: Dear reviewer, we have planned this study only for the use of stored wheat in relation to Pakistan. When Government of Pakistan release stored wheat, it is at very low price as compared to corn. However, in the current study, we stored wheat by ourself, not purchased from the government at lower price. Therefore, we did not mention economic data of the current study. If you still believe, economic data should be added than we may use previous price of wheat that was given by Government of Pakistan to Pakistan Poultry Association. However, that data will not reflect the actual situation or future prospective  

Table 3 lacks the data of body weight, only contains weight gain. the initial and final body weight are very important.

Ans: yes, we agree with you that initial and final body weight are important to present any data on performance parameters. However, many published studies have provided only weight gain data. Therefore, we also used only weight gain data in the current study  

Table 6, what did you mean by 0-35 in carcass traits

Ans: we apologies on mistake. We have corrected it

Reviewer 2 Report

This study  evaluates the effects of stored wheat based diet (1.5 and 2.5 years stored wheat) with and without the supplementation of xylanase and phytase enzymes, used in combination or individually, on  performance parameters, digestibility, digesta viscosity and carcass characteristics of broilers.

The results show that  the inclusion in diet of broilers of the exogenous enzymes xylanase and phytase, alone or in combination, enhances their feed intake, their body weight gain, and also improves the feed conversion ratio. In addition, this supplementation enhances the nutrient digestibility, mitigating NSPs effect and reduces the digesta viscosity of broilers.

The manuscript is well structured but to make comprehension and reading easier a grammatical and linguistic revision of the entire text is necessary.

Below the list of the grammar mystakes and typographical errors found in the text and points to be clarified:

Line 13 : add “a” before “staple” and replace “store” with “stores”;

Lines 19; 21: add an “s” to “enzyme”;

Line 20: add “The” before “current study”;

Lines 22; 23: delete “the” before “performance” and before “broilers”;

Line 23: add “s” to “broiler”;

Line 25; 34: add “s” to “year”. Check this form all over the paper.

Line 26: delete “the” before “both”;

Line 39: delete the article “the” before “plant” and add and “s” to the verb “consist”.

Line 40: delete the second “and” and add a full stop.

Line 44: use the plural form of the verbs.

Line 70: the sentence “Better increase in digestion and absorption of nutrients” would be more clear in this way “Better digestion and  increase in absorption of nutrients”.

Line 74: add “with” after the verb “fed”.

Lines 85-86 : add “a“ before “staple”; replace “wheat” with “it” and add a semicolomn after “Pakistan”. Add also “in this respect” and the article “The” before “Government” and use the third person of the verb store.

Line 87: “after a couple of years” is the correct form.

Line 89: Correct the sentence with “we hypothesized that”.

Line 93: delete the article “the” before broiler.

Line 100: add “the” before “shed”.

Line 111: correct the word “formulation”.

Lines 111-112: Maybe you meant this :“The composition of ingredients and their related data used in the formulation of experimental diets were taken from Brazilian Tables for Poultry and Swine [13].” Please correct the wrong forms.

Line 113: delete “was” after “were” and use the plural form “rations”.

Line 117: add “a” before “Chinese”.

Line 163: per maggior chiarezza  delete “the” and add “each”.

Line 194: use the plural form “samples”.

Line 197: replace “were” with “was”.

Lines 228 and 251: add a “with” after the verb “fed” in Table 3;4 captions.

Lines 158-159: “Fecal digesta samples were collected for digestibility assay at day 35. The floor of each pen was covered with plastic sheets to prevent contamination from bedding material.” Can you clarify when the plastic sheets were posed? On day 34?

The tables are appropriate but for an easier interpretation pay attention to the format (especially Table 3). Please explain in the caption the meaning of the small letters a, b, c, d next to the numbers.

Lines 246-248: These lines could be summirized in a comparison with concept expressed in the previous sentence.

Line 288: please use the third person of the verbs.

Line 289: Please use the correct form“Improving” and “is”. Do you mean “stored grains” ?

Line 290: use the correct form “their endogenous enzymes” instead of “of grain’s endogenous enzymes”

Line 291: delete the “and” before “inclusion” and replace it with a full stop. Then add a “the” before “inclusion”. The use of “or barely” is not clear, please clarify

Line 296: add “the” before “inclusion”

Lines 298-300: Do you mean, “At day 21, greater weight gain was observed both in birds fed with 1.5-year-old wheat-based diet contained both xylanase and phytase enzymes in combination and  2.5-year-old wheat-based diet.” ? Please use “greater” instead of “better” also in line 328; add “with” after “fed”.

Line 307: use “having” instead of “have”

Line 318: add “it” before “was”

Lines 322; 328; 352; 381; 383; 385: add “with” after “fed”

Line 328: This sentence needs to be reformulated like this “At day 35, greater weight gain and FCR were observed in the birds fed with both a 1.5-year-old and a 2.5-year-old wheat based  diets  supplemented both with xylanase and phytase.” Please correct it

Lines 332-333: use the third person of the verbs “ reduces” and “releases”

Line 333: Please correct the adverb, it is “similarly to”

Line 338: please use the verb “go parallel”

Line 345;349: Please check and correct all the NSPs in the paper. The plural form sometimes is missing and subsequently the verb is singular.

Line 352-354: Please rephrase the sentences in this way: “Less viscosity was observed in the birds fed with the 100%-2.5 YOW+Xyl+Phy diet and the 100%-352 1.5 YOW+Xyl+Phy one (P<0.05).”

Line 355: add a “;” after the brakets

Lines 369 and 405: Please cite the name of the Author, in the text not only the reference number.

Line 383: add a “;” before “similarly” and delete “and”;

Lines 384-385: use the correct form “similarly to” and “diets”

Lines 386-387: The sentence can be easier to understand in this way “Woyengo, et al. documented that xylanase  improved apparent amino acid digestibility [54]”

Line 400: add “the” before “combination”

Lines 401-405: please rephrase and summarise the sentence.

The manuscript is scientifically sound and the experimental design is appropriate to test the hypothesis formulated and reproducible following the details given in the “Materials and methods” section.

There is no need to repeat the ethical statement at the beginning of the paragraph entitled Materials and Methods.

The cited references are relevant but could be improved referencing to:

-Tiwari SP, Gendley MK, Pathak AK, Gupta R. Influence of an enzyme cocktail and phytase individually or in combination in Ven Cobb broiler chickens. Br Poult Sci. 2010 Feb;51(1):92-100. doi: 10.1080/00071660903457187. PMID: 20390573.

-Cowieson AJ, Singh DN, Adeola O. Prediction of ingredient quality and the effect of a combination of xylanase, amylase, protease and phytase in the diets of broiler chicks. 2. Energy and nutrient utilisation. Br Poult Sci. 2006 Aug;47(4):490-500. doi: 10.1080/00071660600830611. PMID: 16905476.

The results obtained provide an advancement of the current knowledge on useful integration of diet of broilers capable of enhance their performances; indeed this paper makes available, for the first time, data about the effect of stored wheat (for 1.5 year on the one hand and 2,5 years on the other) on viscosity of digesta and digestion of broilers.

Maybe it would have been interesting to discuss more data presented in Table 6.

The conclusions are consistent with the evidence and arguments presented .

Taking all this into account the work fits the journal aims.

Author Response

This study  evaluates the effects of stored wheat based diet (1.5 and 2.5 years stored wheat) with and without the supplementation of xylanase and phytase enzymes, used in combination or individually, on  performance parameters, digestibility, digesta viscosity and carcass characteristics of broilers.

The results show that  the inclusion in diet of broilers of the exogenous enzymes xylanase and phytase, alone or in combination, enhances their feed intake, their body weight gain, and also improves the feed conversion ratio. In addition, this supplementation enhances the nutrient digestibility, mitigating NSPs effect and reduces the digesta viscosity of broilers.

The manuscript is well structured but to make comprehension and reading easier a grammatical and linguistic revision of the entire text is necessary.

Dear Reviewer,

I really appreciate your effort and time to review our manuscript. I would like to thank for your positive comments and compliment. We have tired our best to improve manuscript according to your suggestions and we mainly focused on grammatical and linguistic of manuscript during revision. We also got help from friend to improve the grammatical and linguistic of manuscript and I hope you will find manuscript suitable for publication this time

Below the list of the grammar mystakes and typographical errors found in the text and points to be clarified:

Line 13 : add “a” before “staple” and replace “store” with “stores”;

Ans: Thanks for suggestion. We have revised the sentence as suggested by you. Now the revised sentence seemsWheat is a staple food in Pakistan and Government of Pakistan stores huge corrected as per reviewer’s suggestion….’ Please see line 14 of the revised manuscript

Lines 19; 21: add an “s” to “enzyme”;

Ans: we appreciate your comment. We have corrected as per your suggestion. Now the corrected sentence seems ‘This study will provide the evidence that storage of wheat will  require exogenous enzymes in the diet ….’ Please check line 19-20 of the revised manuscript

Line 20: add “The” before “current study”;

Ans: Thanks for suggestion. We have incorporated your suggestion and now the revised sentence seemsThe current study was conducted to evaluate the effects of  stored……………’ please check line 21 of the revised manuscript

Lines 22; 23: delete “the” before “performance” and before “broilers”;

Ans: deleted as per reviewer’s suggestion

Line 23: add “s” to “broiler”;

“s” has been added to “broiler” revised text seems ‘For this purpose, a total of 640 day old male broilers ………………..’ please check line 24 of the revised manuscript

Line 25; 34: add “s” to “year”. Check this form all over the paper.

Ans: now it is “years”, corrected all over the paper

Line 26: delete “the” before “both”;

‘the” have been deleted before “both” and now the revised sentence seems ‘In the current study, experimental feeds were prepared by supplementing exogenous enzymes in both basal diets with xylanase…’ please check line 27 of the revised manuscript

Line 39: delete the article “the” before “plant” and add and “s” to the verb “consist”.

Ans: deleted the article “the” before “plant” and added “s” to the verb “consist, now it is “consists”. Now the revised sentence seems ‘Poultry diet mainly consists of plant-based ingredients that …..’please check line 40 of the revised manuscript

Line 40: delete the second “and” and add a full stop.

Ans: deleted the second “and” and added a full stop as per reviewer’s suggestion. Now the revised sentence seems ‘Non-starch polysaccharides (NSPs) are the most common anti-nutritional factors present in plant-based ingredients diet’ please check line 41 of the revised manuscript

Line 44: use the plural form of the verbs.

Ans: used the plural form of the verbs as per reviewer’s suggestion

Line 70: the sentence “Better increase in digestion and absorption of nutrients” would be more clear in this way “Better digestion and  increase in absorption of nutrients”.

Ans: Thanks a lot for nice suggestion. We have revised the sentence and new sentence seemsBetter digestion and increase in absorption of nutrients result in …..’ please check line 71-72 of the revised manuscript

Line 74: add “with” after the verb “fed”.

Ans: added “with” after the verb “fed” as per reviewer’s suggestion

Lines 85-86 : add “a“ before “staple”; replace “wheat” with “it” and add a semicolomn after “Pakistan”. Add also “in this respect” and the article “The” before “Government” and use the third person of the verb store.

Ans: we apologies on mistake. We have corrected it. Now the corrected sentence seems ‘This study is of particular interest in the countries where wheat is a staple food, and it is stored to maintain food supply chain especially during scarcity period like in Pakistan; and in this respect the Government of Pakistan stored huge amount of wheat every year’ please check the revised manuscript.

Line 87: “after a couple of years” is the correct form.

Ans: thanks for suggestion. We have corrected it as suggested by you. Please check the line 88 of the revised manuscript. Now the revised sentence seemsAfter a couple of years, surplus stored wheat is provided to poultry feed …….’

Line 89: Correct the sentence with “we hypothesized that”.

Ans: corrected as per reviewer’s suggestion. Now the revised sentence seems ‘We hypothesized that storage of wheat for 1.5 and 2.5 ….’ Please check the line 90 of the revised manuscirpt

Line 93: delete the article “the” before broiler.

Ans: deleted the article “the” before broiler as per reviewer’s suggestion

Line 100: add “the” before “shed”.

Ans: added “the” before “shed” as per reviewer’s suggestion

Line 111: correct the word “formulation”.

Ans: correct the word “formulation” as per reviewer’s suggestion

Lines 111-112: Maybe you meant this :“The composition of ingredients and their related data used in the formulation of experimental diets were taken from Brazilian Tables for Poultry and Swine [13].” Please correct the wrong forms.

Ans: yes, you are right.  We have corrected as suggested by you. Now the revised sentence seems ‘The composition of ingredient and their related data used in the formulation of experimental diets were taken from Brazilian Tables for Poultry and Swine’ please check line 108-109 of the revised manuscript

Line 113: delete “was” after “were” and use the plural form “rations”.

Ans: deleted “was” after “were” and used the plural form “rations” in the revised manuscript

Line 117: add “a” before “Chinese”.

Ans: added “a” before “Chinese. Please check line 115 of the revised manuscript

Line 163: per maggior chiarezza  delete “the” and add “each”.

Ans: thanks for suggestions. We have corrected the sentence as per your suggestions. Now the revised sentence seemsAsh was estimated by burning each sample in muffle furnace …’ please check line 159 of the revised manuscript

Line 194: use the plural form “samples”.

Ans: used the plural form as per your suggestion

Line 197: replace “were” with “was”.

Ans: replaced as “were” with “was”.

Lines 228 and 251: add a “with” after the verb “fed” in Table 3;4 captions.

Ans: added “with” after the verb “fed” in Table 3;4 captions. Thanks for nice suggestion

Lines 158-159: “Fecal digesta samples were collected for digestibility assay at day 35. The floor of each pen was covered with plastic sheets to prevent contamination from bedding material.” Can you clarify when the plastic sheets were posed? On day 34?

Ans: added “on day 34” to clarify when the plastic sheets were posed

The tables are appropriate but for an easier interpretation pay attention to the format (especially Table 3). Please explain in the caption the meaning of the small letters a, b, c, d next to the numbers.

Ans: format of tables has been changed as other reviewer asked to change the statistical analysis.

Lines 246-248: These lines could be summirized in a comparison with concept expressed in the previous sentence.

Ans: thanks for nice suggestion. We have summarized it. Please check line 252-253 of the revised manuscript. The revised sentence seems ‘these diets without the supplementation of xylanase and phytase had significantly lowered CP, CF, EE and ash digestibility (P<0.05’

Line 288: please use the third person of the verbs.

Ans: used the third person of the verbs in this sentence of the revised manuscript. Please check the discussion first sentence

Line 289: Please use the correct form “Improving” and “is”. Do you mean “stored grains” ?

Ans: we apologies on the mistake. We have corrected it, yes meant stored grains.

Line 290: use the correct form “their endogenous enzymes” instead of “of grain’s endogenous enzymes”

Ans: sorry on this mistake. The new sentence seemsImproving nutritional value of stored grains are due to activation of their endogenous enzymes especially phytase and NSPs degrading enzymes in barley and wheat’ please check line 290-292 of the revised manuscript

Line 291: delete the “and” before “inclusion” and replace it with a full stop. Then add a “the” before “inclusion”. The use of “or barely” is not clear, please clarify

Ans: corrected as per reviewers’ suggestion.

Line 296: add “the” before “inclusion”

Ans: added

Lines 298-300: Do you mean, “At day 21, greater weight gain was observed both in birds fed with 1.5-year-old wheat-based diet contained both xylanase and phytase enzymes in combination and  2.5-year-old wheat-based diet.” ? Please use “greater” instead of “better” also in line 328; add “with” after “fed”.

Ans: I am really sorry on this mistake. Yes, I mean ;At day 21, greater weight gain was observed both in birds fed with 1.5-year-old wheat-based diet contained both xylanase and phytase enzymes in combination and  2.5-year-old wheat-based diet’  and we have corrected as per your suggestion

Line 307: use “having” instead of “have”

Ans: used “having” instead of “have

Line 318: add “it” before “was”

Ans: added “it” before “was”

Lines 322; 328; 352; 381; 383; 385: add “with” after “fed”

Ans: added “with” after “fed” in these lines

Line 328: This sentence needs to be reformulated like this “At day 35, greater weight gain and FCR were observed in the birds fed with both a 1.5-year-old and a 2.5-year-old wheat based  diets  supplemented both with xylanase and phytase.” Please correct it

Ans: corrected as per reviewer’s suggestion. Please check line 329-331 of the revised manuscript

Lines 332-333: use the third person of the verbs “ reduces” and “releases”

Ans: used the third person of the verbs “ reduced” and “released”

Line 333: Please correct the adverb, it is “similarly to”

Ans: corrected

Line 338: please use the verb “go parallel”

Ans: used the verb “go parallel”

Line 345;349: Please check and correct all the NSPs in the paper. The plural form sometimes is missing and subsequently the verb is singular.

Ans: we apologies on this mistake. We have corrected it in all manuscript.

Line 352-354: Please rephrase the sentences in this way: “Less viscosity was observed in the birds fed with the 100%-2.5 YOW+Xyl+Phy diet and the 100%-352 1.5 YOW+Xyl+Phy one (P<0.05).”

Ans: thanks a lot for suggestion. We have revised the sentence as suggested by you. Please check the revised manuscript line 253-254.

Line 355: add a “;” after the brackets

Ans: added “;” as per reviewer’s suggestion

Lines 369 and 405: Please cite the name of the Author, in the text not only the reference number.

Ans:  cited the names of authors in reference 27 and 57

Line 383: add a “;” before “similarly” and delete “and”;

Ans:  corrected as per reviewer’s suggestion

Lines 384-385: use the correct form “similarly to” and “diets”

Ans: corrected as per reviewer’s suggestion

Lines 386-387: The sentence can be easier to understand in this way “Woyengo, et al. documented that xylanase  improved apparent amino acid digestibility [54]”

Ans: corrected as per reviewer’s suggestion

Line 400: add “the” before “combination”

Ans: added

Lines 401-405: please rephrase and summarise the sentence.

Ans: thanks dear reviewer for your suggestion. We have rephrased and summarized the sentence as suggested by you. Now the revised sentence seems ‘In the present study, combined effects of phytase and xylanase on crude fiber digestibility were significant and enhanced crude fiber digestibility could be the result of degradation of the hemicellulose of crude fiber by exogenous xylanase, as reported by Pettersson and Aman, [59]’ please check line 402-405 of the revised manuscript

The manuscript is scientifically sound and the experimental design is appropriate to test the hypothesis formulated and reproducible following the details given in the “Materials and methods” section.

There is no need to repeat the ethical statement at the beginning of the paragraph entitled Materials and Methods.

Ans: Thanks a lot for your suggestions, efforts, and your time to review our manuscript. As for as ethical statement, it was prerequisite during submission. We will try to remove it in final draft as per instruction of journal and editor.

Q: The cited references are relevant but could be improved referencing to:

-Tiwari SP, Gendley MK, Pathak AK, Gupta R. Influence of an enzyme cocktail and phytase individually or in combination in Ven Cobb broiler chickens. Br Poult Sci. 2010 Feb;51(1):92-100. doi: 10.1080/00071660903457187. PMID: 20390573.

-Cowieson AJ, Singh DN, Adeola O. Prediction of ingredient quality and the effect of a combination of xylanase, amylase, protease and phytase in the diets of broiler chicks. 2. Energy and nutrient utilisation. Br Poult Sci. 2006 Aug;47(4):490-500. doi: 10.1080/00071660600830611. PMID: 16905476.

Ans: Respected Reviewer, we have added the reference in the revised manuscript at appropriate place. Thanks for suggestions and your hard work

The results obtained provide an advancement of the current knowledge on useful integration of diet of broilers capable of enhance their performances; indeed this paper makes available, for the first time, data about the effect of stored wheat (for 1.5 year on the one hand and 2,5 years on the other) on viscosity of digesta and digestion of broilers.

Ans: Thanks a lot to appreciate our efforts. Indeed your suggestions had greatly improved the manuscript.

Maybe it would have been interesting to discuss more data presented in Table 6.

Ans: We have tried to rewrite some results and discussion as suggested by other reviewer too

The conclusions are consistent with the evidence and arguments presented .

Ans: Thanks a lot

Taking all this into account the work fits the journal aims.

Ans: Once again, I would like to thanks for your hard work and time to review this manuscript. I really appreciate your effort and sincerity to improve this manuscript. Your suggestions were really wonderful that had really improved our manuscript

Round 2

Reviewer 1 Report

Thanks for authors responses, good job